# Comparison between Automated and Manual Detection of Lava Fountains from Fixed Monitoring Thermal Cameras at Etna Volcano, Italy

**Sonia Calvari** [1,*] and **Giuseppe Nunnari** [2]

[1]  Istituto Nazionale di Geofisica e Vulcanologia, Osservatorio Etneo—Sezione di Catania, Piazza Roma 2, 95125 Catania, Italy
[2]  Dipartimento di Ingegneria Elettrica, Elettronica e Informatica, Università degli Studi di Catania, Viale A. Doria 6, 95122 Catania, Italy; giuseppe.nunnari@dieei.unict.it
*  Correspondence: sonia.calvari@ingv.it

**Abstract:** The Etna volcano is renowned worldwide for its extraordinary lava fountains that rise several kilometers above the vent and feed eruptive columns, then drift hundreds of kilometers away from the source. The Italian Istituto Nazionale di Geofisica e Vulcanologia-Osservatorio Etneo (INGV-OE) is responsible for the monitoring of Mt. Etna, and for this reason, has deployed a network of visible and thermal cameras around the volcano. From these cameras, INGV-OE keeps a keen eye, and is able to observe the eruptive activity, promptly advising the civil protection and aviation authorities of any changes, as well as quantifying the spread of lava flows and the extent of pyroclastic and ash plumes by using a careful analysis of the videos recorded by the monitoring cameras. However, most of the work involves analysis carried out by hand, which is necessarily approximate and time-consuming, thus limiting the usefulness of these results for a prompt hazard assessment. In addition, the start of lava fountains is often a gradual process, increasing in strength from Strombolian activity, to intermediate explosive activity, and eventually leading to sustained lava fountains. The thresholds between these different fields (Strombolian, Intermediate, and lava fountains) are not clear cut, and are often very difficult to distinguish by a manual analysis of the images. In this paper, we presented an automated routine that, when applied to thermal images and with good weather conditions, allowed us to detect (1) the starting and ending time of each lava fountain, (2) the area occupied by hot pyroclasts, (3) the elevation reached by the lava fountains over time, and (4) eventually, to calculate in real-time the erupted volume of pyroclasts, giving results close to the manual analysis but more focused on the sustained portion of the lava fountain, which is also the most dangerous. This routine can also be applied to other active volcanoes, allowing a prompt and uniform definition of the timing of the lava fountain eruptive activity, as well as the magnitude and intensity of the event.

**Keywords:** automated detection; remote sensing; lava fountains; Etna volcano

## 1. Introduction

New data and interpretations have emerged of the geodynamics of the eastern Sicily point to Etna as a volcano, undergoing an evolutionary phase where a future increase in highly energetic explosive activity is possible [1]. As a matter of fact, the last three decades of Etna's activity were characterized by frequent highly explosive eruptions, here called paroxysms [2,3]. Paroxysms at Etna are characterized by lava fountaining lasting 1–2 h, reaching the height of 1–3 km above the crater, and generating conspicuous and lengthy ash plumes that can drift hundreds of kilometers from the vents [4,5], often accompanied by short-lasting lava overflows from the crater rim [6,7]. During the last few decades, Etna volcano underwent several eruptions characterized by lava flows mainly from the summit vents, alternating with short-lasting but powerful explosive episodes [2,3,8–10]. In

particular, between 2011 and 2015, Etna produced more than 50 such eruptions [3,10–12], releasing a cumulative erupted volume of a similar order to a major flank eruption [2], which was normally ten times greater than summit activity [13]. Given that explosive paroxysms can have a major impact on aviation [14], on road and traffic conditions, and also on the villages on the slope of the volcano [15–17], it is of paramount importance for a volcano observatory such as the Istituto Nazionale di Geofisica e Vulcanologia-Osservatorio Etneo (INGV-OE) to be able to raise an early warning as soon as possible, and then to advise the civil protection authorities of its possible impact on human activities [18–21]. The first and most important parameter to be detected as soon as possible is the timing of start and end of any impending activity, and this information needs to be completed with the extent of the ash plume, lava flows and lava fountains, and with the volume erupted [3,6,8,12,22,23]. The volume erupted during a lava fountain (LF) episode quantifies the magnitude of the event, whereas the eruption rate determines its intensity [24]. The information gathered from the monitoring system is then used to inform the civil protection of the magnitude and intensity of the event, and also in models routinely used for the prediction of the extent and distribution of the eruption products, which, at Etna, often comprise both lava flows [25,26] and pyroclastics [27–29]. An automated procedure to map the lava flows from the images of the thermal monitoring cameras was recently developed [6,7], whereas the LF detection and the estimation of the erupted volumes were normally carried out by manual analysis of the images [3,8,30]. The pyroclastic volume estimated by thermal images was compared to the total erupted volume estimated by strain and with the lava flow volume erupted during each episode as estimated by satellite [30], and an error of ~20% was estimated for the calculated fluid volume, comprising gas plus pyroclastics [31].

A sustained LF normally gives rise to an eruptive column comprising three main portions: a lower and innermost zone called gas-thrust region with the highest velocity at the exit of the erupting vent; an intermediate zone, where convective movements of the hot mixture of gas and tephra allow ingestion of the cold surrounding air, thereby slowing down the spreading hot jet, and an uppermost zone reaching the buoyancy zone and consisting of the laterally spreading umbrella region [32,33]. In addition, the eruptive columns at Etna can be distinguished into weak plumes if bent in the wind direction due to wind speeds greater than 10 m s$^{-1}$, or strong to intermediate plumes when rising vertically above the vent or slightly bent in the wind direction for wind speeds lower than 10 m s$^{-1}$ [3].

LF heights were determined following several different methods. One of the earliest, applied at Kilauea, was carried out by digitizing film from 8 mm time-lapse movie cameras deployed on the ground, and using a few theodolite measurements as calibration points for the film [34]. At Etna, a C++ code was developed in order to explore the INGV-OE thermal image library for image processing [35], applying appropriate thresholds, and converting the color images into a binary black and white image over which the maximum vertical extent of the LF can be easily retrieved. A less automated but sometimes more precise system involves the visual and manual analysis of each image [3,8], given that the thermal images can be affected by low gas, weather or ash clouds [36–40] that may reduce the automated measurement of the LF height, or the LF jet may be inclined [3,8,30,35]. Calibrated images of visible cameras can also be used to estimate the vertical extent of proximal ash plumes associated with the lava fountains [12,22].

One of the most difficult challenges in volcanology is to determine when an eruption is over, especially when it includes multiple episodes and long pauses [41], although sometimes, a gradual decline of the mass eruption rate may anticipate the end of the eruption [42]. During the lava fountain activity at Etna, the start and ending time, as well as any early warning alarm, is given on the basis of the volcanic tremor and infrasound [18,43–46]. However, a volcanic tremor does not allow us to calculate the volume erupted [3,8], and does not provide information about the extension of the lava fountains and ash plume [5,22]. Another useful device is the borehole strainmeter, which allows cal-

culating the total erupted volume [47,48], comprising both lava flows and pyroclastics [30]. However, in order to assess the impact on the population, the amount of solely the pyroclastics component erupted during a lava fountain event needs to be established, because this affects the stability of roofs, the cleaning up of roads and motorways, the impact on the nearby Catania international airport, and the health effects on the local population [49,50]. Automated routines for volcanic activity detection and characterization have been recently developed [23,45,51,52], and they will probably resolve most of the issues related with early warning alarms. In this paper, we presented a new automated routine that, when applied to the images recorded by the thermal monitoring cameras, allowed us to calculate (1) lava fountain height, (2) area of the lava fountain jet, and (3) volume of the erupted tephra, using the formula applied by Calvari et al. [3,8]. We compared these results with those obtained by a manual analysis, discussing limits and advantages, and future possible improvements.

## 2. Methods

The INGV-OE thermal camera network for monitoring Etna volcano includes four fixed, continuously operating thermal cameras located on the flanks of the volcano at different distances from and looking towards the summit (Figure 1, Table 1). The images recorded by these cameras are transmitted to the INGV-OE Operative Room and displayed in real-time to allow continuous monitoring of the volcano. From this perspective, operators have the task of recognizing the type of event as early as possible. Thermal cameras are remote-sensing ground-based fixed devices that have significantly improved INGV-OE's observational capabilities. They allowed us to monitor the summit area continuously, and to identify and locate eruptive events. Only poor weather conditions, and especially thick clouds of water vapor, gas, and/or volcanic ash [37–40] may affect the visibility of the cameras, and consequently, the reliability of the acquired images, by partially or totally hiding what is happening in the monitored area. When, by manual examination of images, we recognized that visibility was limited to a few frames or interfering clouds were low and only partially obscured the lava fountain (LF), a linear interpolation was carried out on the data [3]. The manual analysis of the thermal images followed what was described by Calvari et al. [3,8].

**Table 1.** List of the INGV monitoring thermal cameras used in this paper and their main features. The field of view is considered at the crater rim.

| Label | Type | Location | Distance from the Craters (km) | Frame Rate | Field of View |
|-------|------|----------|--------------------------------|------------|---------------|
| ENT | Thermal FLIR A40M | Nicolosi, South flank 730 m a.s.l. | 15.0 | 2 frames/s | 320 × 240 pixels |
| EBT | Thermal FLIR A320 | Bronte, NW flank 85 m a.s.l. | 13.5 | 2 frames/s | 25° × 18.8° |
| EMCT | Thermal FLIR A320 | Mt. Cagliato, East flank 1160 m a.s.l. | 8.3 | 2 frames/s | 320 × 240 pixels |
| EMOT | Thermal FLIR A320 | Montagnola, South flank 2600 m a.s.l. | 3.0 | 1 frame/s | 320 × 240 pixels |

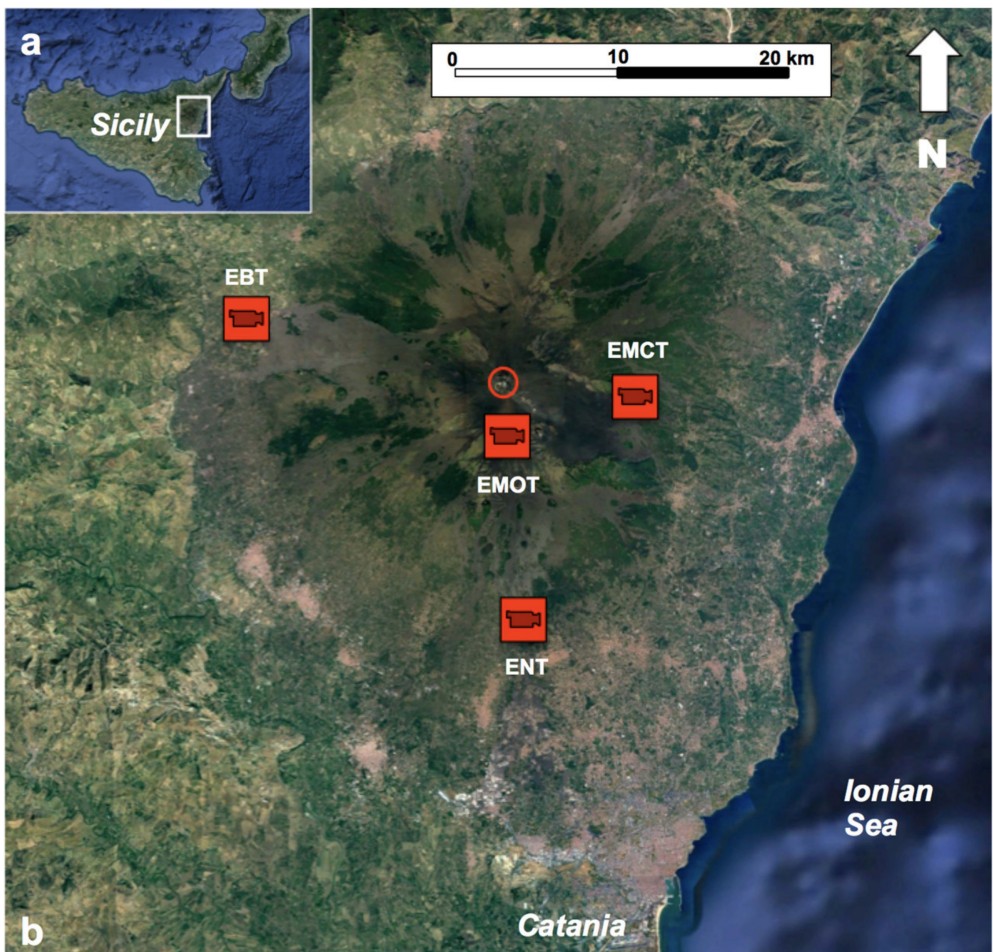

**Figure 1.** (**a**) Google map of Sicily, with the white rectangle showing the area magnified in (**b**), which is the Google map of Etna volcano showing the position of the INGV thermal monitoring cameras used in this study. The red circle indicates the position of the summit craters producing the lava fountain activity.

A sequence of LF episodes occurred at Mt. Etna between 2020 and 2022, and the list of these events is given in Table 2. We have analyzed the images of the thermal cameras that recorded the event (Figure 1, Table 2), and chose for each episode the one offering the best view and the entire vertical extension of the LF, as a function of the LF size, wind direction, and consequent ash plume fallout. In cases of rotating ash plumes, we used the integration of images from more than one camera. The manual analysis of the camera images allowed retrieving the starting and ending time of each episode, the duration expressed in minutes and seconds, the maximum height of the lava fountain and its average value, as well as the erupted volume of pyroclastics and the time-averaged discharge rate (TADR, [53]). The volume of pyroclastics was calculated by following the method by Calvari et al. [3,8], based on the measurements of the LF heights at time lapses of 1 min, and considering a constant vent radius of 15 m and a pyroclastic ratio of 0.18% of the total erupted fluids comprising gas plus pyroclastics. All these results, obtained from the manual analysis of the thermal camera images, are reported in Table 2.

**Table 2.** Paroxysmal explosive episodes occurring at Etna volcano between 2020 and 2022 (updated to 23 February 2022). The first column reports the episode's progressive number. All times are expressed in UTC. LF = lava fountain. TADR = time averaged discharge rate [53]. Data from episode # 32 are lacking because there was no visibility from any of the thermal monitoring cameras.

| Ep. # | Date | Starting Time (hh:mm) | Ending Time (hh:mm) | Duration (in Minutes and in Seconds) | Label of Cam-era Used | Max LF Height (m above the Crater) | Mean LF Height (m above the Crater) | LF Volume (×10$^6$ m$^3$) | TADR (m$^3$ s$^{-1}$) |
|---|---|---|---|---|---|---|---|---|---|
| 1 | 13 December 2020 | 22:00 | 22:48 | 48 min. 2880 s | ENT | 514 | 231 | 0.24 | 84 |
| 2 | 13–14 December 2020 | 23:58 | 00:11 | 13 min. 780 s | ENT | 400 | 235 | 0.07 | 90 |
| 3 | 14 December 2020 | 01:02 | 01:40 | 38 min. 2280 s | ENT | 286 | 121 | 0.13 | 59 |
| 4 | 21 December 2020 | 09:11 | 09:59 | 48 min. 2880 s | EBT | 3080 | 1296 | 0.58 | 201 |
| 5 | 22 December 2020 | 03:05 | 05:13 | 128 min. 7680 s | ENT | 800 | 295 | 0.72 | 93 |
| 6 | 18 January 2021 | 19:38 | 21:03 | 85 min. 5100 s | ENT | 1067 | 343 | 0.52 | 101 |
| 7 | 16 February 2021 | 16:11 | 17:02 | 51 min. 3060 s | EMCT | 1560 | 757 | 0.46 | 150 |
| 8 | 17–18 February 2021 | 22:32 | 00:51 | 139 min. 8340 s | EMCT | 1230 | 358 | 0.82 | 98 |
| 9 | 19 February 2021 | 08:16 | 10:06 | 110 min. 6600 s | EMCT | 1365 | 492 | 0.78 | 118 |
| 10 | 20–21 February 2021 | 21:32 | 01:15 | 223 min. 13,380 s | EMCT | 1500 | 386 | 1.43 | 107 |
| 11 | 22–23 February 2021 | 21:17 | 00:03 | 166 min. 9960 s | ENT EMOT EMCT | 3667 | 686 | 1.19 | 120 |
| 12 | 23 February 2021 | 03:52 | 04:50 | 58 min. 3480 s | ENT EMOT EMCT | 900 | 337 | 0.32 | 92 |
| 13 | 24 February 2021 | 18:56 | 21:41 | 165 min. 9900 s | ENT | 1800 | 649 | 1.37 | 139 |
| 14 | 28 February 2021 | 07:31 | 08:34 | 63 min. 3780 s | ENT | 3600 | 1376 | 0.70 | 185 |
| 15 | 2 March 2021 | 11:23 | 14:50 | 207 min. 12,420 s | EMOT | 606 | 278 | 1.03 | 83 |
| 16 | 4 March 2021 | 01:30 | 04:10 | 160 min. 9600 s | ENT | 600 | 204 | 0.75 | 78 |

**Table 2.** *Cont.*

| Ep. # | Date | Starting Time (hh:mm) | Ending Time (hh:mm) | Duration (in Minutes and in Seconds) | Label of Cam-era Used | Max LF Height (m above the Crater) | Mean LF Height (m above the Crater) | LF Volume ($\times 10^6$ m$^3$) | TADR (m$^3$ s$^{-1}$) |
|---|---|---|---|---|---|---|---|---|---|
| 17 | 4 March 2021 | 07:11 | 09:32 | 141 min. 8460 s | ENT | 3233 | 1275 | 1.58 | 186 |
| 18 | 7 March 2021 | 04:10 | 07:01 | 171 min. 10,260 s | EMOT EBT | 4000 | 638 | 1.07 | 104 |
| 19 | 9–10 March 2021 | 23:55 | 02:46 | 171 min. 10,260 s | ENT EMCT | 1860 | 655 | 1.44 | 140 |
| 20 | 12 March 2021 | 07:35 | 09:45 | 130 min. 7800 s | ENT EBT | 2400 | 1149 | 1.63 | 209 |
| 21 | 14–15 March 2021 | 23:12 | 01:42 | 150 min. 9000 s | ENT | 1333 | 670 | 0.53 | 59 |
| 22 | 17 March 2021 | 01:30 | 04:57 | 207 min. 12,420 s | ENT | 1533 | 538 | 1.58 | 128 |
| 23 | 19 March 2021 | 08:18 | 10:13 | 115 min. 6900 s | EMOT | 629 | 171 | 0.75 | 108 |
| 24 | 23–24 March 2021 | 21:33 | 08:19 | 646 min. 38,760 s | EMCT | 1333 | 456 | 4.56 | 118 |
| 25 | 31 March– 1 April 2021 | 19:30 | 08:53 | 803 min. 48,180 s | EMCT | 630 | 241 | 4.10 | 85 |
| 26 | 19 May 2021 | 00:50 | 04:25 | 215 min. 12,900 s | ENT | 667 | 482 | 1.59 | 124 |
| 27 | 21 May 2021 | 00:50 | 02:44 | 114 min. 6840 s | EMCT | 1533 | 683 | 0.99 | 145 |
| 28 | 22 May 2021 | 20:27 | 22:08 | 101 min. 6060 s | ENT | 1200 | 649 | 0.87 | 143 |
| 29 | 24 May 2021 | 20:25 | 21:49 | 84 min. 5040 s | ENT | 1467 | 831 | 0.82 | 162 |
| 30 | 25 May 2021 | 18:20 | 18:53 | 33 min. 1980 s | ENT | 533 | 317 | 0.20 | 102 |
| 31 | 26 May 2021 | 10:20 | 11:10 | 50 min. 3000 s | ENT | 1267 | 627 | 0.41 | 137 |
| 32 | 27 May 2021 | 12:00 | 13:00 | 60 min. 3600 s | EMCT | Poor visibility | Poor visibility | Poor visibility | Poor visibility |
| 33 | 28 May 2021 | 06:30 | 07:27 | 57 min. 3420 s | ENT | 800 | 433 | 0.40 | 116 |
| 34 | 28 May 2021 | 16:05 | 16:11 | 6 min. 360 s | ENT | 333 | 295 | 0.04 | 112 |

**Table 2.** *Cont.*

| Ep. # | Date | Starting Time (hh:mm) | Ending Time (hh:mm) | Duration (in Minutes and in Seconds) | Label of Cam-era Used | Max LF Height (m above the Crater) | Mean LF Height (m above the Crater) | LF Volume ($\times 10^6$ m$^3$) | TADR (m$^3$ s$^{-1}$) |
|---|---|---|---|---|---|---|---|---|---|
| 35 | 28 May 2021 | 19:48 | 20:50 | 62 min. 3720 s | ENT | 1000 | 601 | 0.51 | 138 |
| 36 | 30 May 2021 | 04:20 | 05:44 | 84 min. 5040 s | ENT | 1000 | 589 | 0.69 | 137 |
| 37 | 2 June 2021 | 08:30 | 10:46 | 136 min. 8160 s | EBT | 1640 | 924 | 1.38 | 170 |
| 38 | 4 June 2021 | 16:12 | 17:40 | 88 min. 5280 s | EMCT | 1170 | 665 | 0.76 | 143 |
| 39 | 12 June 2021 | 20:00 | 21:46 | 106 min. 6360 s | EMCT | 810 | 438 | 0.73 | 115 |
| 40 | 14 June 2021 | 21:15 | 22:21 | 66 min. 3960 s | EMCT | 870 | 419 | 0.45 | 112 |
| 41 | 16 June 2021 | 11:37 | 12:38 | 61 min. 3660 s | ENT | 1733 | 673 | 0.52 | 142 |
| 42 | 17 June 2021 | 22:40 | 23:55 | 75 min. 4500 s | EMCT | 1140 | 404 | 0.50 | 111 |
| 43 | 19 June 2021 | 18:47 | 19:35 | 48 min. 2880 s | EMCT | 1140 | 572 | 0.38 | 131 |
| 44 | 20 June 2021 | 22:40 | 23:44 | 64 min. 3840 s | ENT | 2467 | 892 | 0.63 | 165 |
| 45 | 22 June 2021 | 02:30 | 03:45 | 75 min. 4500 s | ENT | 2000 | 848 | 0.71 | 159 |
| 46 | 23 June 2021 | 02:44 | 03:19 | 35 min. 2100 s | ENT | 1867 | 1035 | 0.38 | 183 |
| 47 | 23 June 2021 | 18:00 | 19:12 | 72 min. 4320 s | ENT | 2933 | 1301 | 0.60 | 153 |
| 48 | 24 June 2021 | 09:45 | 10:26 | 41 min. 2460 s | ENT EMOT | 1733 | 825 | 0.39 | 157 |
| 49 | 25 June 2021 | 00:38 | 01:48 | 70 min. 4200 s | ENT | 2333 | 876 | 0.64 | 153 |
| 50 | 25 June 2021 | 18:40 | 19:20 | 40 min. 2400 s | ENT | 1133 | 691 | 0.36 | 149 |
| 51 | 26 June 2021 | 16:00 | 16:38 | 38 min. 2280 s | ENT | 1600 | 772 | 0.35 | 155 |
| 52 | 27 June 2021 | 08:53 | 09:43 | 50 min. 3000 s | ENT | 1600 | 674 | 0.43 | 143 |

**Table 2.** *Cont.*

| Ep. # | Date | Starting Time (hh:mm) | Ending Time (hh:mm) | Duration (in Minutes and in Seconds) | Label of Cam-era Used | Max LF Height (m above the Crater) | Mean LF Height (m above the Crater) | LF Volume ($\times 10^6$ m$^3$) | TADR (m$^3$ s$^{-1}$) |
|---|---|---|---|---|---|---|---|---|---|
| 53 | 28 June 2021 | 14:25 | 15:30 | 65 min. 3900 s | EBT | 2390 | 1211 | 0.75 | 193 |
| 54 | 1–2 July 2021 | 22:50 | 00:27 | 97 min. 5820 s | ENT | 1800 | 804 | 0.91 | 156 |
| 55 | 4 July 2021 | 15:15 | 16:50 | 95 min. 5700 s | ENT | 1467 | 873 | 0.94 | 164 |
| 56 | 6 July 2021 | 22:16 | 23:44 | 88 min. 5280 s | EBT | 3270 | 1673 | 1.20 | 227 |
| 57 | 8 July 2021 | 20:35 | 22:12 | 97 min. 5820 s | EBT | 2710 | 1242 | 1.12 | 192 |
| 58 | 14 July 2021 | 10:45 | 12:30 | 105 min. 6300 s | EBT | 2230 | 1097 | 1.15 | 183 |
| 59 | 20 July 2021 | 05:10 | 08:11 | 181 min. 10,860 s | EBT | 3510 | 1533 | 2.26 | 208 |
| 60 | 31 July 2021 | 19:44 | 23:37 | 233 min. 13,980 s | EBT | 3830 | 1473 | 2.83 | 202 |
| 61 | 9 August 2021 | 02:07 | 04:11 | 124 min. 7440 s | EBT | 2390 | 1280 | 1.48 | 199 |
| 62 | 29 August 2021 | 16:24 | 17:55 | 91 min. 5460 s | EBT | 2390 | 1310 | 1.10 | 202 |
| 63 | 21 September 2021 | 07:21 | 08:35 | 74 min. 4440 s | ENT | 2333 | 1234 | 0.88 | 199 |
| 64 | 23 October 2021 | 08:20 | 10:17 | 117 min. 7020 s | ENT | 4000 | 1844 | 1.63 | 232 |
| 65 | 10 February 2022 | 18:40 | 21:56 | 196 min. 11,760 s | ENT | 5714 | 2160 | 2.88 | 245 |
| 66 | 21 February 2022 | 11:11 | 12:50 | 99 min. 5940 s | ENT | 4057 | 1865 | 1.39 | 234 |
| | | | | Average Duration (min./s) | | Average Max LF height (m) | Average Mean LF height (m) | Average LF volume ($\times 10^6$ m$^3$) | Average TADR (m$^3$ s$^{-1}$) |
| | | | | 120/7171 | | 1815 | 784 | 0.993 | 144.75 |

The dataset consisted of 66 episodes of LF recorded between 13 December 2020 and 21 February 2022 at Mt Etna by using the four thermal cameras of the INGV-OE monitoring network listed in Table 1, and whose position is shown in Figure 1. Additional technical details on the thermal sensors can be found in Calvari et al. [3,29]. Original data were provided as *.avi* format files, each containing about 5 min of recorded volcanic activity. Information concerning the name of the camera and the starting time of each file were embedded in its name. For instance, the filename *EMOT 20210319–082500.avi*, refers to a file recorded by the camera named EMOT, starting on 19 March 2021 at time 08:25:00. All times are UTC. Other information, such as the duration, the frame-rate, and other useful video properties, such as width, height, bits per pixel, and the video format were embedded in the file object. Files were pre-processed in order to crop the color bar and the information about the acquisition time and camera name, which are normally embedded in the frame.

In order to detect the presence of LF, each frame was converted from the original RGB format to grayscale and further binarized by adopting a threshold luminance value, specified as a value in the range [0, 1]. In this way, the hottest objects, such as newly erupted or cooling down products, will be represented in the binarized image as white areas, while all the others will be represented in black (Figure 2). Thus, in the absence of hot objects, such as new ejected volcanic matter, hot spots or cooling lava, the binarized images will result completely black. However, as mentioned above, it should be noted that hot objects may not be detected due to the presence of a thick cloud cover. Clearly, the choice of the threshold parameter played a crucial role. In fact, although on one hand, it allowed filtering unwanted information, due, for example, to warming of the monitored area by the Sun, on the other hand, it could also remove useful information, especially in the phase of emergence of an LF episode. Unfortunately, there were no optimal criteria for the choice of this parameter which was therefore made for each camera, adopting the traditional trial and error method. Each binarized image was processed, and detected objects, represented in white color in Figure 2c, were measured in order to obtain:

- the areas $A_i$, $i = 1, N$, $A_i$ being the area in pixels of the i-th object and $N$ the number of recognized objects;
- the coordinates $(x_i, y_i)$ of the centroid, of the i_th object, $x_i$ and $y_i$ being the horizontal and vertical coordinates, respectively.

The overall process, starting from the original image (Figure 2a), through the grayscale (Figure 2b), the binarized (Figure 2c), and finally, to the labeled image (Figure 2d) is reported, as an example, in Figure 2. In particular, the centroids of detected hot objects are shown by the red asterisks in Figure 2d.

The presence of multiple objects, even from an individual LF episode, was due to the fact that the volume occupied by an LF does not have a uniform temperature, as can be seen from Figure 2a where the original colors of the thermal image indicate different temperatures in a scale starting from blue (0 °C) to white (>100 °C). However, for practical reasons, as different hot objects belong to the same individual LF, it is a good practice to consider all detected objects as a single one. In our case, this was obtained by summing up the areas of all detected objects, and calculating the coordinates of a single centroid by a weighted average of the coordinates of the individual centroids, as expressed in Equation (1).

$$x = \sum_{i=1}^{N} w_i x_i; y = \sum_{i=1}^{N} w_i y_i, \ w_i = \frac{A_i}{\sum_{i=1}^{N} A_i} \tag{1}$$

where $A_i$ is the area of the $i$-th object, $N$ is the number of detected objects, and $w_i$ is the normalized area of the $i$-th object. It is straightforward to say that, with considerable approximation, due to the fact that the LF occurs in 3-D volume, while images refer to a 2-D area, the estimated area $A$ is, in some way, related to the volume of hot matter, while the $y$-coordinate of its centroid may be related with the mean elevation. Of course, $A$ and its centroids' coordinates ($x,y$), originally expressed in pixel units, can be converted into geographical units by appropriate conversion constants, depending on the position of the

considered camera with respect to the monitored area. The graphical representation of the area *A* (Figure 3a) and its mean altitude (Figure 3b) for an LF episode occurring on Mt. Etna on 24 February 2021, is shown in Figure 3.

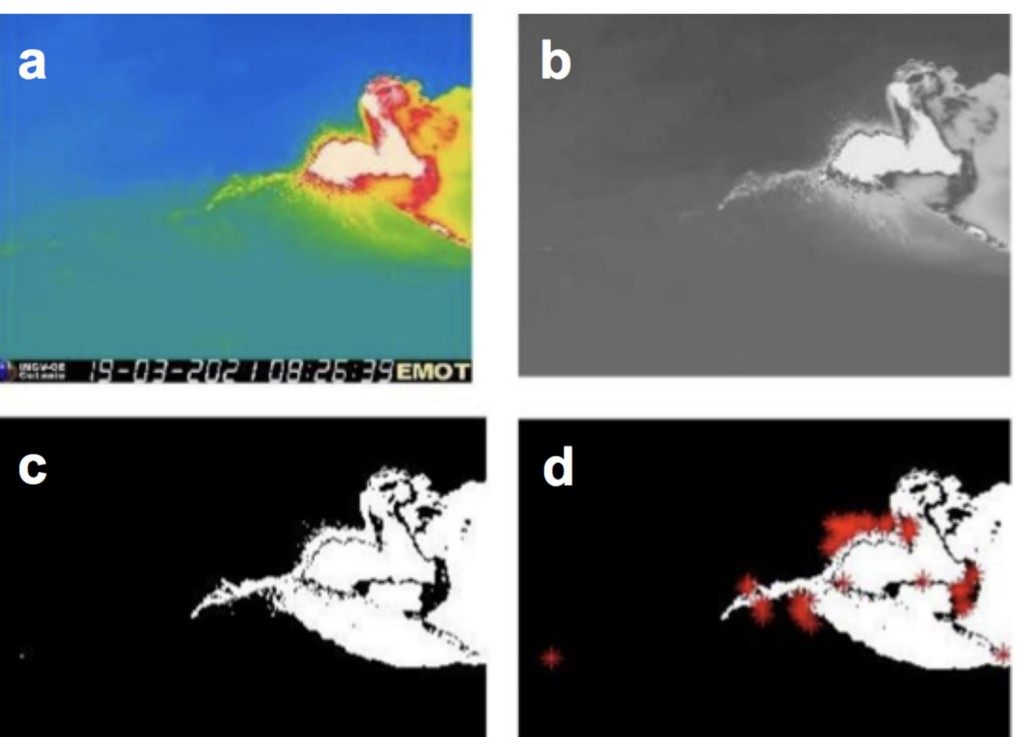

**Figure 2.** (**a**) The original RGB thermal image from the EMOT camera, with temperatures in °C comprised between blue (0 °C) and white (>100 °C); (**b**) the cropped grayscale; (**c**) the binarized image, and (**d**) the binarized image with computed centroids (red asterisks), representing the center of mass of the hot objects detected by the threshold process.

Figure 3 can be interpreted as follows: during 24 February 2021, a mass of previously erupted material was cooling down around the crater area. This can be deduced from the fact that the *y*-coordinate of the centroids (Figure 3b) fluctuates around zero during the time interval from 00:00 to about 19:25. In Figure 3b, for convenience, the zero value of the mean altitude has been arbitrarily set to the average value of the *y*-coordinate, measured throughout the recording period. The lack of signal in the mean altitude (in the reported example, for instance, around 14:20 and 14:40) is due to the thick cloud cover which prevented viewing of the cooling mass. After 19:15, an LF appeared, as can be seen from both the area signal (Figure 3a) and the increasing mean altitude of the center of mass (Figure 3b). The LF continued until about 20:00, after which, the erupted material began a cooling process. It should be observed that while the value of the area slowly decreased, the value of the mean altitude of the hot objects almost instantaneously decreased when the LF ended. As can be seen from Figure 3, an LF can therefore be recognized by the characteristic bell-shaped distribution of both the area and mean altitude time series. This suggests a criterion for identifying the time mark to be associated with the beginning and the end of the paroxysmal phenomenon, by using a Change Point Detection (CPD) algorithm, which will be explained in Section 3.2.

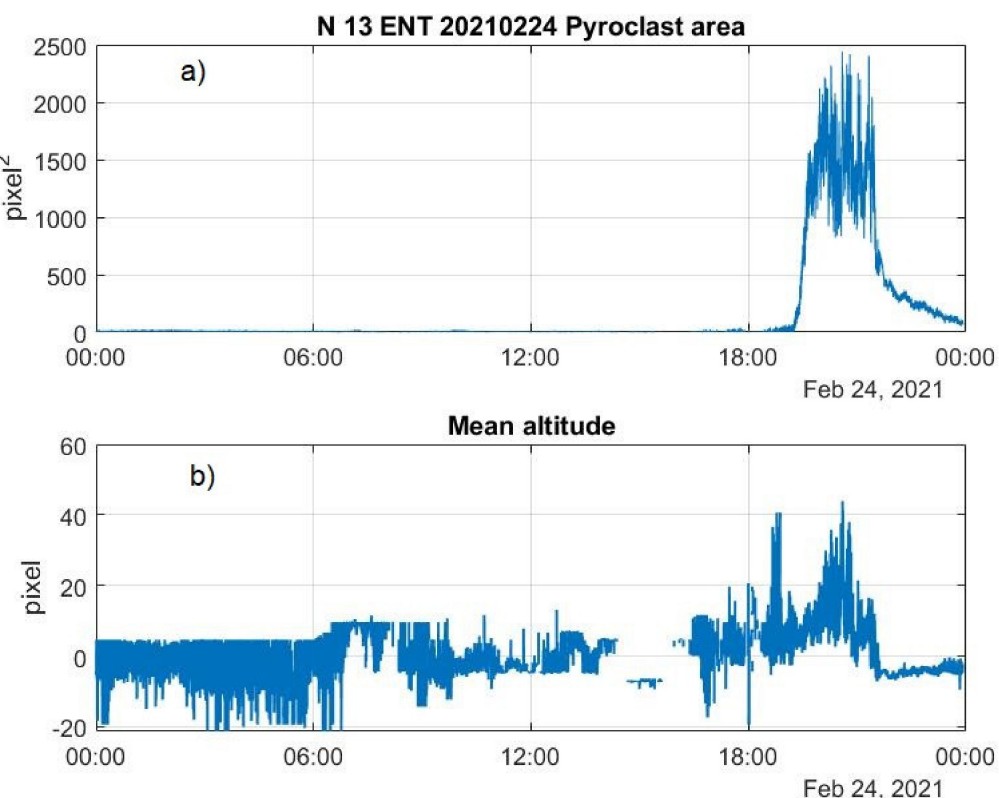

**Figure 3.** Area (**a**) and mean altitude (**b**) of the lava fountain episode at Mt Etna on 24 February 2021.

## 3. Results

### 3.1. Manual Estimation of the Eruptive Activity

Of the 66 LF episodes that occurred between 13 December 2020 and 21 February 2022, only one could not be detected because of poor weather conditions: the episode of 27 May 2021, when there was poor visibility from all the monitoring cameras. In all the other cases, visibility was more than 80% of the duration of the episode. Thus, for the short lapses of time when there was no visibility, the LF heights were linearly interpolated. The total volume of erupted pyroclastics during the 65 episodes was ~65 × 10$^6$ m$^3$. The average duration of the 65 episodes listed in Table 2 was of 120 min (minimum 6 min, maximum 803 min), or 7171 s (minimum 360 s, maximum 48,180 s); the average erupted volume of pyroclastics was 0.99 × 10$^6$ m$^3$, with 41 × 10$^3$ m$^3$ as minimum value and 4.6 × 10$^6$ m$^3$ as maximum value; the TADR, calculated for the only pyroclastic portion of the episodes and for the whole duration of each paroxysmal event, was 145 m$^3$ s$^{-1}$ on average, spanning from a minimum of 59 m$^3$ s$^{-1}$ and a maximum of 245 m$^3$ s$^{-1}$; maximum LF heights above the vents were 1815 m on average, spanning from a peak value of 5714 m to a minimum of 333 m, and the average heights of the LFs were between 115 m and 778 m, with a peak value of 2160. All values are listed in Table 2.

From a volcanological point of view, it is worth noting that if we exclude the two outliers of 23 and 31 March 2021, which emitted more than 4 × 10$^6$ m$^3$ of tephra, the volume of pyroclastics which erupted during the LF episodes occurring between mid-December 2020 and February 2022 increased with time. This can clearly be seen from Figure 4, where we report the distribution of the erupted volumes with time and its linear trend. In addition, the time-averaged discharge rate (TADR) and its trend increased with time (Figure 5). These observations, i.e., of the increase of erupted volume of pyroclastics and rate of eruption with time, suggested that the sequence of paroxysmal events was not yet over [31].

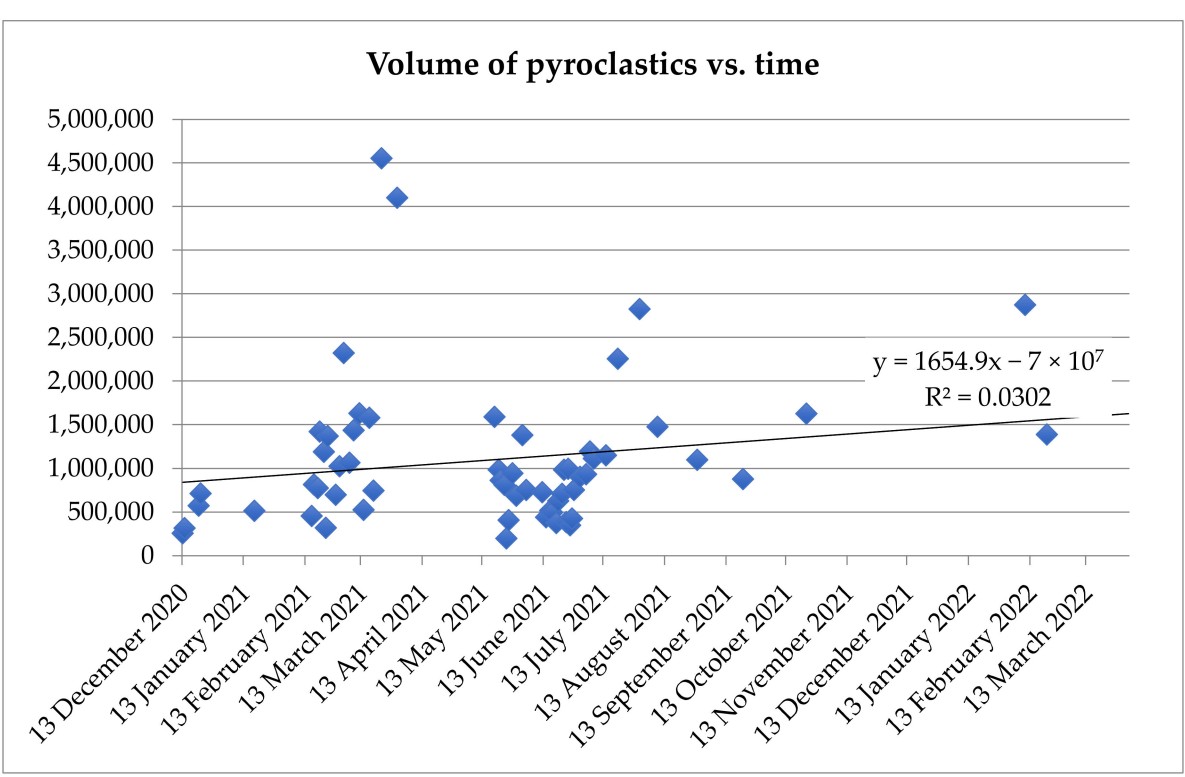

**Figure 4.** Volume of pyroclastics (*y*-axis, expressed in m$^3$) against time (*x*-axis, date) for the 65 explosive episodes listed in Table 2, together with a linear trend and its formula.

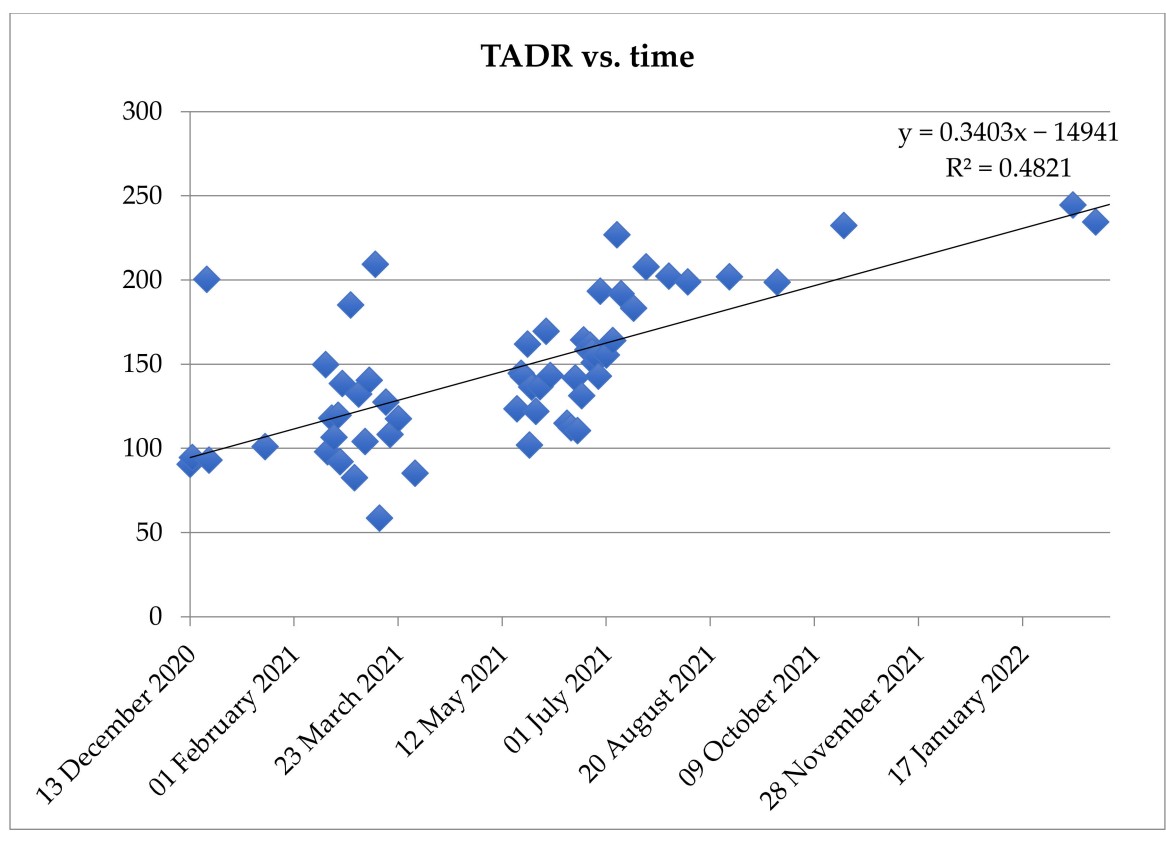

**Figure 5.** Time-averaged discharge rate of pyroclastics (TADR, *y*-axis, expressed in m$^3$ s$^{-1}$) erupted during the lava fountain episodes against time (*x*-axis, date). The graph displays the data for the 65 explosive episodes listed in Table 2, together with a linear trend and its formula.

*3.2. Change Point Detection*

A change point represents a transition between different states in a process that generates the time series. Change point detection (CPD) can be defined as the problem of choosing between two alternatives: no change or indeed, a change occurred. CPD algorithms are traditionally classified as online or offline [54]. Offline algorithms consider the whole data set at once and try to recognize where the change occurred. Thus, the aim in this case, is to identify all the sequence change points in batch mode. In contrast, online, or in real-time, algorithms run concurrently with the process they are monitoring, processing each data point as it becomes available, with the goal of detecting a change point as soon as possible after it occurs, ideally before the next data point arrives. In practice, no CPD algorithm operates in perfect real-time because it must wait for new data before determining if a change point occurred. However, different online algorithms require different amounts of new data before a change point can be detected. Based on this observation, an online algorithm, which needs at least $\varepsilon$ samples in the new batch of data to be able to find a change, is usually denoted as $\varepsilon$-real time. Therefore, offline algorithms can be viewed as $\infty$-real time whereas the best online algorithm is 1-real time, because for every data point, it can predict whether or not a change point occurs before the new data point. Smaller $\varepsilon$ values may lead to stronger, prompter CPD algorithms. To find a change point in a time series, a global optimization approach can be used with the following basic algorithm:

1.  Choose a point and divide the signal into two sections.
2.  Compute an empirical estimate of the desired statistical property for each section.
3.  At each point within a section, measure how much the property deviates from the empirical estimate, and at the end, add the deviation for all points.
4.  Add the deviations section-to-section to find the total residual error.
5.  Vary the location of the division point until the total residual error attains a minimum.

As noted above, the search for a change point $k$ can be formulated as an optimization problem where the cost function $J(k)$ to minimize it can be written, in the general case as:

$$J(k) = \sum_{i=1}^{k-1} \Delta(x_i; \chi([x_1, \ldots, x_{k-1}])) + \sum_{i=k}^{N} \Delta(x_i; \chi([x_k, \ldots, x_N])) \tag{2}$$

where $\{x_1, x_2, \ldots x_N\}$ is the time series, $\chi$ is the chosen statistic, and $\Delta$ is the deviation measurement. In particular, when $\chi$ is the mean, the cost function assumes the following form:

$$J(k) = \sum_{i=1}^{k-1} \left(x_i - \langle x \rangle_1^{k-1}\right)^2 + \sum_{i=k}^{N} \left(x_i - \langle x \rangle_k^N\right)^2 \tag{3}$$

where the symbol $\langle \cdot \rangle$ indicates the mean operator.

Another aspect to be considered, when formulating the optimization problem, is that signals of practical interest have more than one change point. Furthermore, the number of change points $K$ is often not known a priori. To handle these features, the cost function can be generalized as:

$$J(k) = \sum_{r}^{K-1} \sum_{i=k_r}^{k_{r+1}-1} \Delta(x_i; \chi([x_k, \ldots, x_{r+1}])) + \beta K \tag{4}$$

where $k_0$ and $k_K$ are, respectively, the indexes of the first and the last sample of the signal. In the expression (4), the term $\beta K$ is a penalty term, linearly increasing with the number of change points $K$, which avoids the problem of overfitting [55]. Here, $\beta$ represents a positive coefficient that weights the number of searched change points. Indeed, in an extreme case (i.e., $\beta = 0$), $J(K)$ reaches the minimum value (i.e., 0) when every point becomes a change point (i.e., $K = N$).

The algorithm described above for a univariate time series, can easily be extended to the case of a multivariate time series, which was the case, for instance, of a data set recorded by a GPS network [54]. In this case, the cost function was evaluated, of course, over the

whole set of available time series. The software considered in this work was implemented in MATLAB based on the CPD algorithms described in [55,56]. The package can help the user at various levels. The lowest level is to consider the software to obtain the time series of area *A* and mean altitude *MA* of detected hot objects from images of volcanic activity, following the algorithm described in the previous section. Subsequently, the user, based on a visual inspection of these time series, can indicate presumed times for the starting and ending times of the LF, and request the software to calculate other quantities of interest such as the volumes emitted. Another possibility is to leave the software to search for the transition times from Strombolian to paroxysmal activity using the CPD algorithm, possibly selecting the statistic to be used to perform the detection (i.e., abrupt changes in the mean, in the variance or in the slope). While the latter possibility is preferable when the time series shows clear LF episodes, the former is more suitable when the automated detection of the transition from Strombolian to paroxysmal activity and vice versa can be more problematic due to external noise sources (e.g., poor visibility or interference by other kinds of hot objects).

### 3.3. Timing the Lava Fountains Occurring at Etna during 2020–2022

The main advantages of performing a computer process analysis of LF images are the following:

- the user can quickly analyze the content of the image files recorded over days, an operation which, carried out manually, requires a considerable amount of time;
- the user can speed up the computation of key quantities such as height and duration of the LF, which are necessary for the calculation of the volumes of erupted material);
- it is possible to implement algorithms for automatically timing the transition from Strombolian to paroxysmal activity, which is otherwise left to human judgment, gaining in uniformity and repeatability;
- in case of lack of visibility, since it is necessary to proceed with interpolation of the data, a rather difficult operation to perform manually, the user can resort to automated interpolation techniques (e.g., linear interpolation, nearest, etc.).

However, it should not be overlooked that quantitative measurements of the LF parameters, starting from the images, require overcoming several non-negligible difficulties. First of all, the aforementioned lack of visibility can make the reliability of the measurements poor. In fact, interpolation techniques can help in cases of limited amounts of data, but obviously cannot replace them when significant amounts are missing.

Furthermore, since the LFs are recognized as hot objects, they can be confused with hot objects of other kinds. Hot areas are very often formed due to the sunlight reflection of both the ground (Figure 6a) and the clouds (Figure 6c). At other times, the Sun itself was included in the images as it travels its natural orbit (Figure 6d,e). Other kinds of hot objects, which could be confused with LFs, were cooling lava flows (Figure 6b). Moreover, different hot objects can combine their effects with those of the LFs (Figure 6e,f). Some of these effects could be avoided by using special cameras, but this is not always possible. Since the current state of development of the software does not allow a reliable distinction between the noise and the LF signal, the user, for the purpose of determining the start and end times, can limit the search space, so as to exclude particularly noisy periods.

Normally, the area signal is smoother than the mean altitude signal and therefore, usually when determining the start and end time of an LF using the CPD algorithm, it is preferable to consider the time series of the areas. However, in some cases, it may be useful to consider the series of mean altitude, as described in the following example. For instance, consider the area signal shown in Figure 7a, which refers to an LF occurring on Etna on 20 February 2021. As can be seen, due to the presence of a cooling lava flow, the area signal slowly decreases (thermal hysteresis), making it difficult to accurately perform the automated detection of the ending time of this LF episode. However, this shortcoming can be overcome by performing the CPD timing on the mean altitude signal, which is not affected by the thermal hysteresis (see Figure 7b).

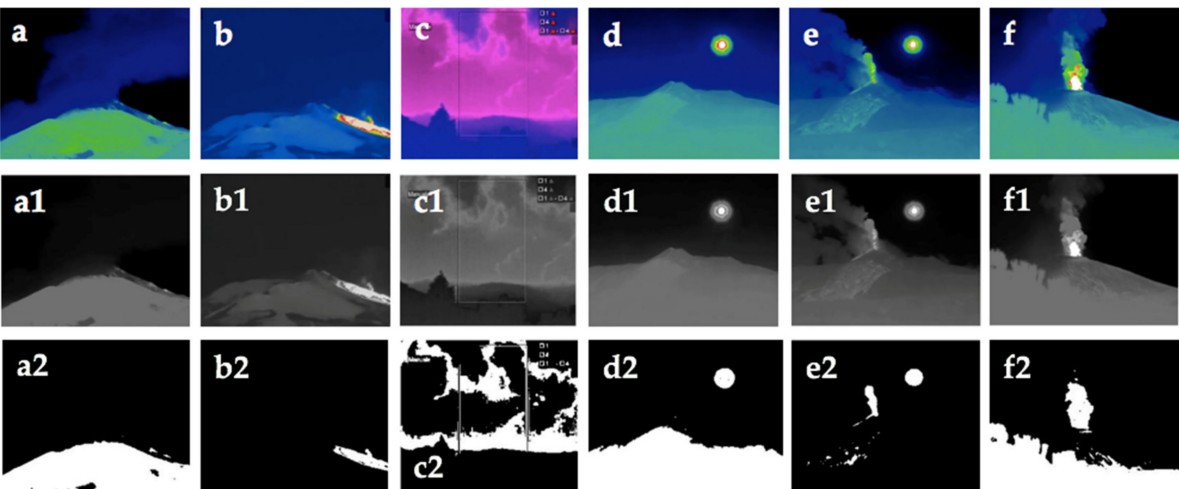

**Figure 6.** (**a**) Effect of the Sun reflection on the ground slope (green area), EMOT camera; (**b**) cooling lava flow (white area), EMOT camera; (**c**) Effect of the Sun on the clouds (pink area), ENT camera; (**d**) the Sun in the camera field of view (white circle), EMCT camera; (**e**) Combined effects: lava fountain in presence of the Sun in the field of view, EMCT camera; (**f**) lava fountain and Sun reflection on the vegetation in the foreground, EBT camera. (**a1**–**f1**) are the corresponding gray images, and (**a2**–**f2**) are the corresponding black and white binarized images.

Here, the term 'timing' will be used to indicate the estimation of the starting and ending time of an LF episode. In particular, for this LF, while performing the CPD timing from the area signal (Figure 8a), the end of the LF is estimated to be $t_{end}$ = 06:38, whereas considering the mean altitude signal, the end of the LF is estimated $t_{end}$ = 00:56, which is closer to that manually estimated ($t_{end}$ = 01:15) and reported in Table 2. This difference was caused by the greater curvature that can be seen in Figure 8a, which was determined by the cooling lava flow area.

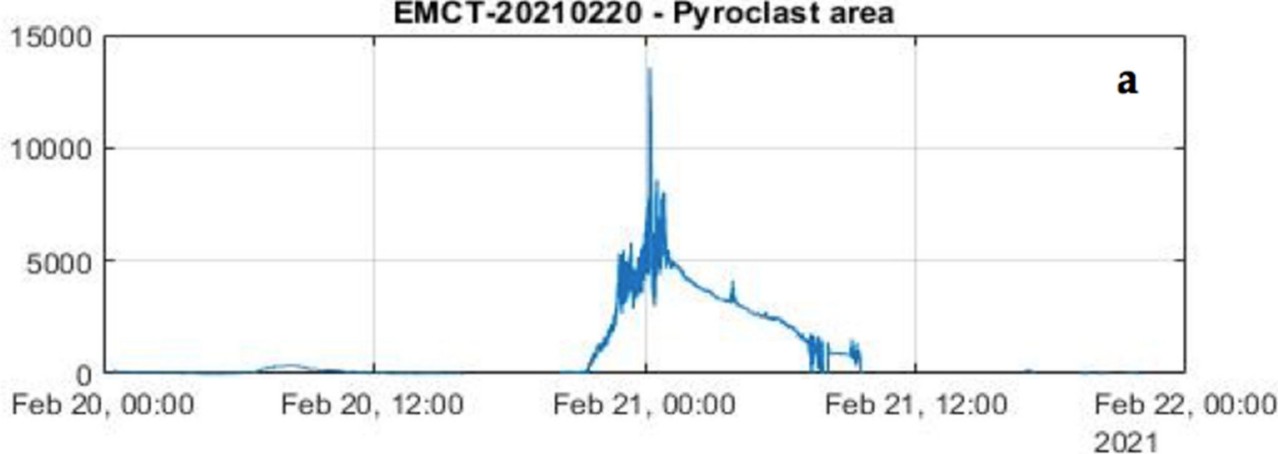

**Figure 7.** *Cont.*

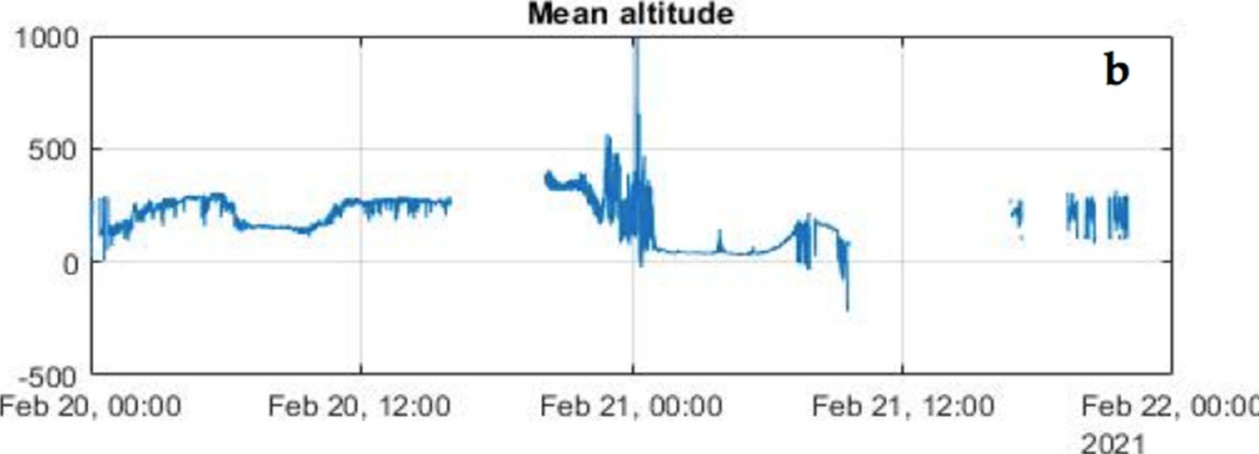

**Figure 7.** (**a**) Area (in pixel²) and (**b**) mean altitude (in pixels above the crater rim) against time of the lava fountain on 20–21 February 2021 at Mt. Etna, retrieved from the EMCT camera.

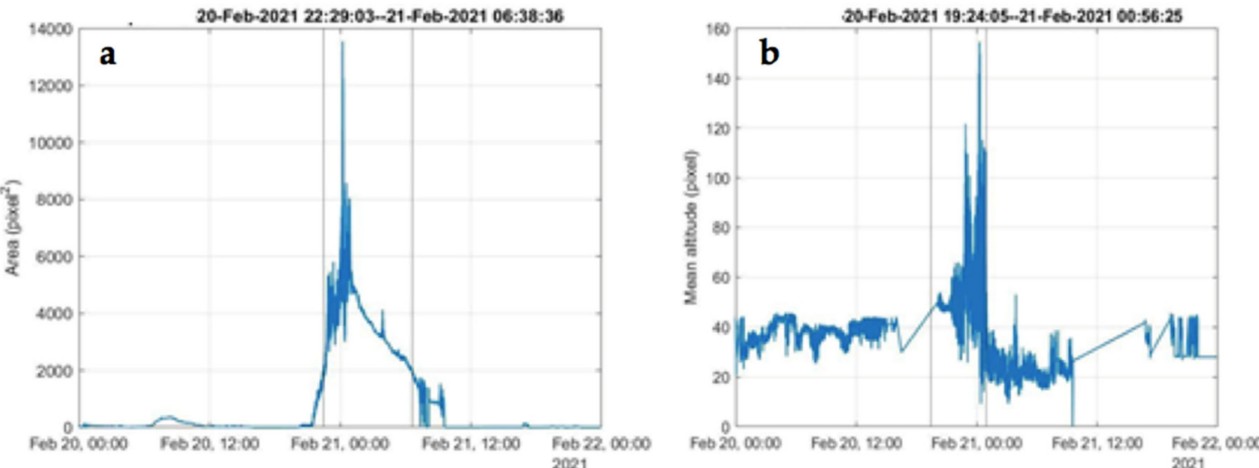

**Figure 8.** (**a**) Area (in pixel²) and (**b**) mean altitude above the crater rim (in pixels, measured above the crater rim) with the timing (gray vertical lines) of start and end of the lava fountain episode. See text for explanation.

### 3.4. Timing the Lava Fountains by a Gaussian Function-Based Approach

In some cases, assuming that a typical LF has a time distribution of area and mean altitude, which roughly has a bell shape, it might be useful to approximate the measured data by using a Gaussian function. This can be useful, for example, when the images are affected by thick clouds passing through the field of view of the camera, generating a trend such as those shown in Figure 9a, which refers to the LF episode on 13 December 2020 and was observed from the ENT camera.

For this LF episode, the manually estimated starting and ending times were 22:00 and 22:48, respectively. However, from Figure 9a, it can be seen that precisely between these two times, the recorded signal is discontinuous due to poor visibility, but nevertheless, it can be seen that the area signal shows a well-detectable peak. Fitting the area samples, it is possible to obtain the results shown in Figure 9b and thus estimate the start and end dates of the LF to be 22:02 and 22:20, respectively, which are closer to the manually assessed corresponding times.

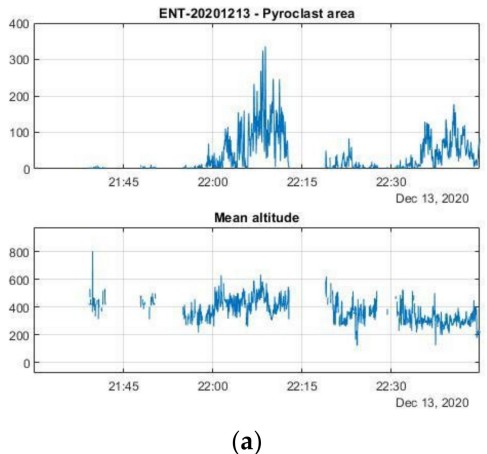
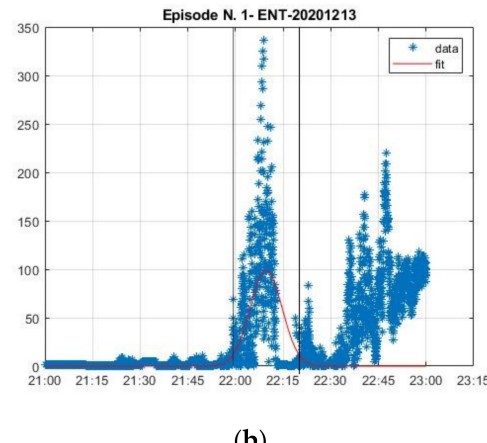

(**a**)
(**b**)

**Figure 9.** (**a**) Area (in pixel$^2$) and mean altitude (in pixels, measured above the crater rim) against time of the 13 December 2020 lava fountain episode as retrieved from the ENT camera, and (**b**) Gaussian interpolation (red line) of the pyroclastic area against time of the 13 December 2020 lava fountain episode retrieved from the ENT camera.

Timing of an LF episode, after having carried out the approximation of the curve by means of a Gaussian function, is simply established by using a threshold approach: the starting time is set as the one in which the recorded data exceed, for the first time, a threshold of the normalized function height. Similarly, the end time is established as the one in which the recorded signal falls, for the first time, below the threshold. In this paper, the threshold value has been set, after a trial and error approach, to be 25% of the maximum value.

Of course, the Gaussian approach can be used for timing LF as an alternative to the CPD one, even when the visibility problems described above do not exist, as shown in the example of Figure 10.

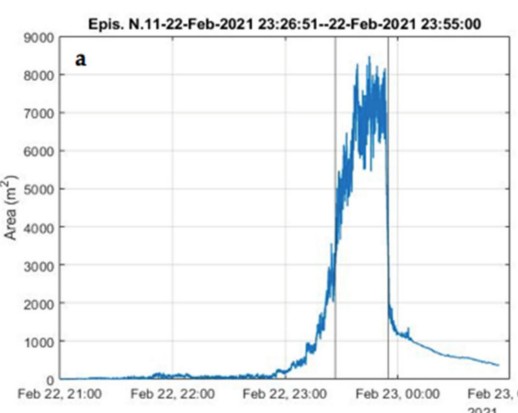
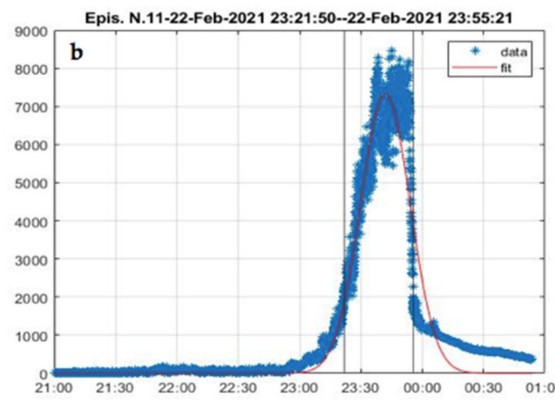

**Figure 10.** Graphs showing the timing of the LF episode N. 11, which occurred on 22–23 February 2021 (Table 2) obtained from (**a**) the CPD method and from (**b**) the Gaussian method. The black vertical lines indicate the start and end time, and the red line in (**b**) is the Gaussian interpolation. Table 2 shows that the manual method indicated the LF episode took place between 21:17 on 22 February and 00:03 of 23 February 2021. In accordance with the manual method, (**a**) shows that the CPD automated timing indicated that the sustained phase of the LF developed between 23:26 and 23:55 on 22 February, and similarly, the Gaussian method (**b**) indicated a timing comprised between 23:21 and 23:55 on 22 February.

The software package presented in the previous sections was considered to perform the timing of the 65 LF episodes in the data set reported in Table 2. Moreover, for each

episode, the heights of the LF at 1 min intervals were used to calculate the total fluid erupted volume from Equations (5) and (6), which included both gas and pyroclastics [8,31]:

$$U = (2gH)^{0.5} \tag{5}$$

$$V = U \cdot A_v \cdot D \tag{6}$$

In expression (5), U is the mean fluid exit velocity at the vent, H is the LF height, and g is the gravity acceleration, while in expression (6), V is the fluid volume (gas + pyroclastic) erupted by the LF, $A_v$ is the vent section area, and D is the duration of the LF in seconds. The vent surface area was calculated by assuming a circular vent with a diameter of 30 m [8] and supposed to be constant. Moreover, we have computed the volume $V_2$ of pyroclastics from the total erupted volume V (gas + pyroclastics), considering 0.18% as the ratio between the volumes of magma and fluids within the eruptive column as typical for Etna's fountains [3].

According to expression (6), the estimated volumes depend on the mean fluid exit velocity U and LF duration D, for assigned values of the vent surface area. Thus, the performances of the automated approach will depend on its readability to estimate height H and duration D of the LF episodes. Concerning the reliability of the automated estimating H, the comparison with the corresponding manual reading, for a few episodes of the data set, is shown in Figure 11.

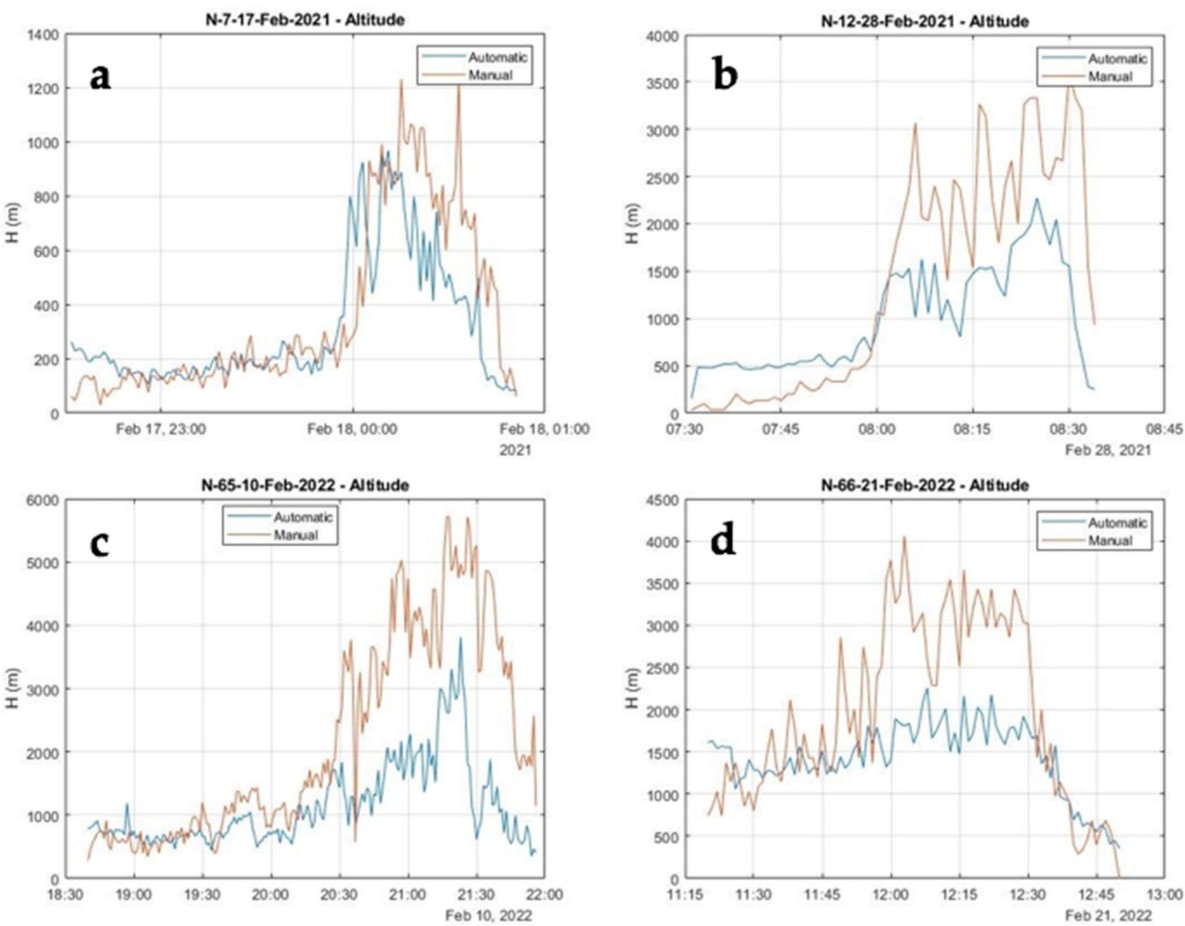

**Figure 11.** Time series of heights estimated by using the automated and the manual approaches for a few episodes of the data set against time, with the blue line for the automated, and the orange line for the manual detection: (**a**) Episode 7, 17–18 February 2021; (**b**) Episode 12, 28 February 2021; (**c**) Episode 65, 10 February 2022, and (**d**) Episode 66, 21 February 2022.

It can be seen in Figure 11 that manual height readings normally have a greater range than those automatically estimated. Here, it should be borne in mind that, as expressed in Equation (1), the automatically measured heights are a weighted average of the centroids, while those measured manually are normally taken as the maximum height of the lava fountain jet taken along the spreading direction. Considering that the heights represent the only geometric element on which the volumes of erupted material depend, it follows that with the automated estimation, these will normally be slightly less when compared to the manual ones, but with the advantage of immediacy. It is also necessary to bear in mind that for the purposes of estimating the volume of erupted matters, it is not so much the precise values of the instantaneous heights that are relevant, but their average value, which therefore also depends on the estimated duration for each individual episode. The comparison among the mean heights for the whole LF episodes after estimating the duration $D$ by using both the CPD and Gaussian approaches are shown in Figure 12. In this figure, the abscissa is the integer $N$ ranging from 1 to 66, i.e., number of LF episodes in the considered dataset (Table 2). The episode 32 is lacking because of poor visibility from all the monitoring cameras (Table 2).

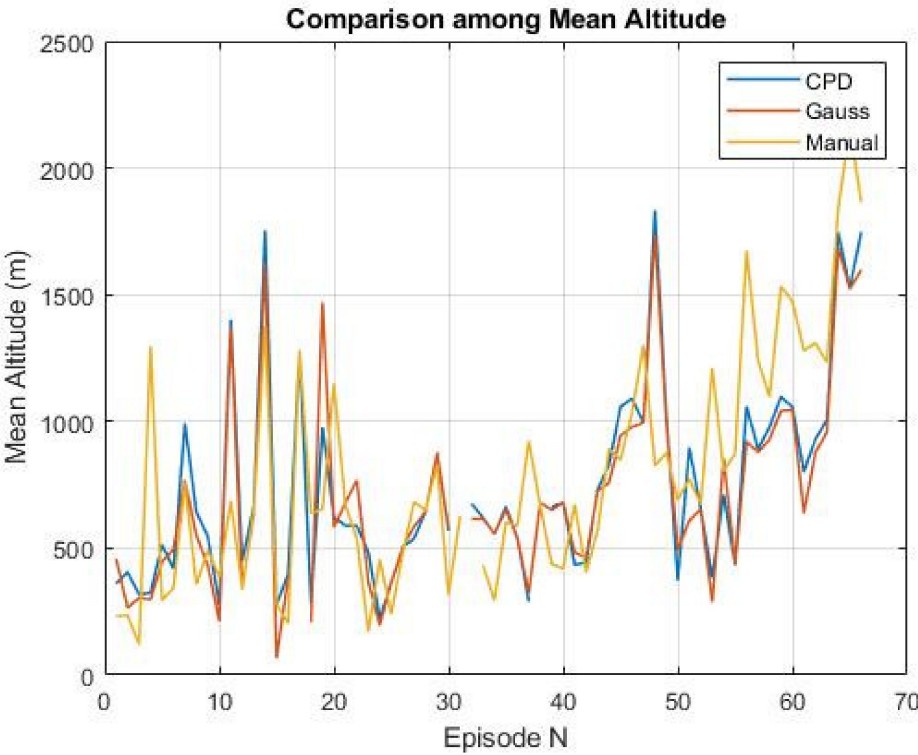

**Figure 12.** Mean altitude for each of the 65 lava fountain episodes estimated after having established the timing of start and end for each recorded time series by using both the CPD (blue line) and the Gaussian (red line) approach. Their values were compared with the corresponding manually estimated values (yellow line). Episode 32 was lacking because of poor visibility from the cameras (see Table 2).

In more detail, Figure 12 shows that for most of the episodes, there is a good agreement between the average mean heights estimated for each episode, not only between those obtained by using the CPD and Gaussian approaches, but also between these and those manually estimated. Altitudes obtained with the manual approach are, on average, 12% lower than those obtained by using the CPD approach, and 7% lower than the Gaussian one.

As regards to the duration $D$ for each episode, the comparison between the automated and the manual estimation is shown in Figure 13.

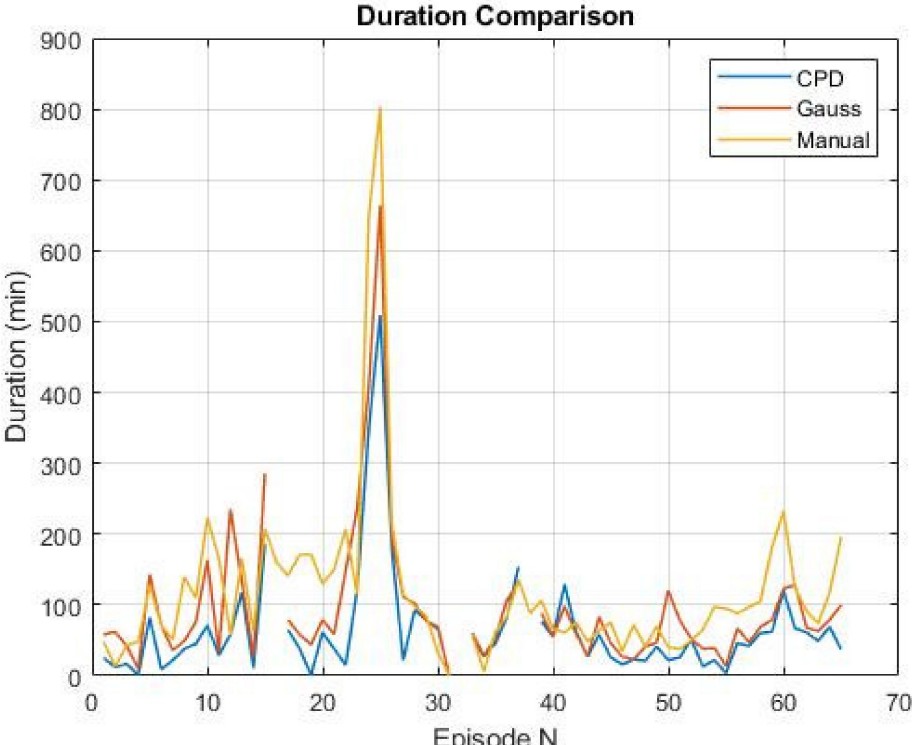

**Figure 13.** Estimated duration performed after timing the LF episodes by the CPD (blue line) and Gaussian (red line) approaches, in comparison with the manual (black line) ones.

Figure 13 shows that the durations obtained by using both the methods for automated timing are generally in good agreement with each other, as well as lower than those obtained manually. In more detail, duration manually estimated was on average about 34% higher, with a standard deviation of 77%, than those estimated by using the CPD, and about 3% higher, with a standard deviation of 93%, than the Gaussian one. To justify the discrepancy, it is worth noting that the automated approaches generally identify the sustained part of each LF, while the manual approach is not able to clearly distinguish the threshold of the intermediate phase preceding the sustained portion of the LF [3,8].

The comparison between estimated volumes and TADR by the three methods are reported in Figure 14a,b. A good agreement between the automated and manual estimation is apparent, bearing in mind that the automated volumes are usually smaller than the manual ones, because the durations refer to the sustained phase of the LF, while the manual and automated TADR are in good agreement because this feature is computed as the ratio between volumes and duration of each LF episode. In more detail, manual estimated volumes are, on average, about 26% higher, with a standard deviation of about 77% than those estimated by using the CPD approach, and 13% lower than the Gaussian, with a standard deviation of about 107%. The TADR manually estimated is about 15% lower that the CPD, with a standard deviation of 77% and 8% lower than the Gaussian, with a standard deviation of 33%.

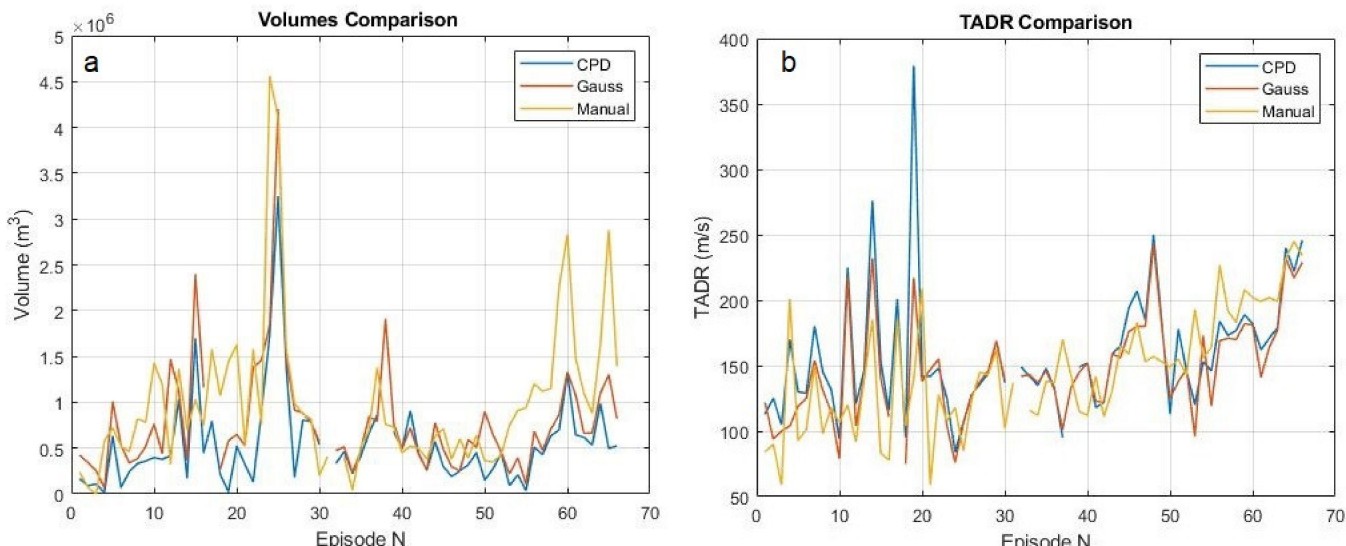

**Figure 14.** (**a**) Comparison among volumes, and (**b**) comparison among TADR obtained from the manual and automated approaches.

Figure 15 shows the differences between the values obtained by the manual and automated routines, and indicates the good agreement between heights of the lava fountain (Figure 15a) and TADR (Figure 15d), and the discrepancies between duration (Figure 15b) and calculated erupted volume of pyroclastics (Figure 15c).

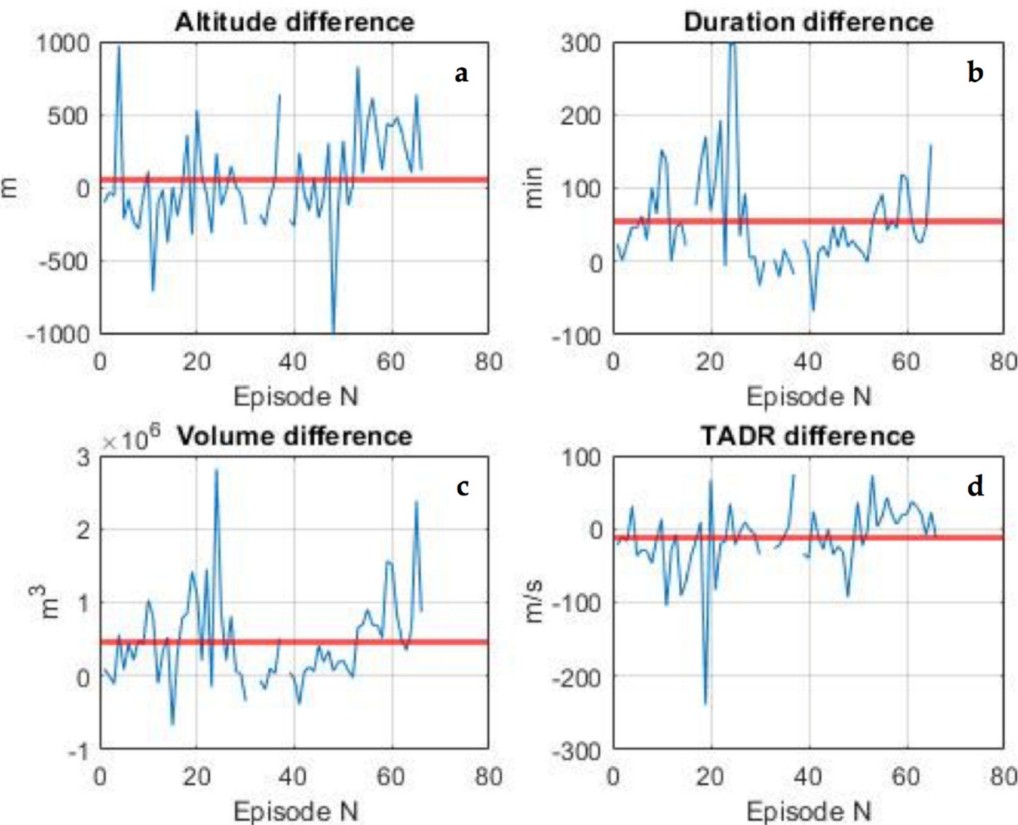

**Figure 15.** Comparison among automated and manual analysis of (**a**) heights of the lava fountain; (**b**) duration of each episode; (**c**) estimated erupted volume of pyroclastics, and (**d**) TADR (time-averaged discharge rate [53]).

Considering the results obtained from the automated routines, the total volume of erupted pyroclastics during the 65 episodes computed by using the automated CPD approach was ~$34.5 \times 10^6$ m$^3$, with a minimum of ~$0.012 \times 10^6$ m$^3$ and a maximum of ~$3.25 \times 10^6$ m$^3$. The average duration of the sustained part of LF was of ~65.7 min (minimum 1 min, maximum 509 min). The TADR, calculated only for the pyroclastic portion of the episodes and for the whole duration of each paroxysmal event, was 159 m$^3$ s$^{-1}$ on average, spanning from a minimum of 84 m$^3$ s$^{-1}$ and a maximum of 379 m$^3$ s$^{-1}$; LF average height above the vents was 745 m, spanning from a peak value of 1834 m to a minimum of 223 m.

## 4. Discussion

In this paper, we have presented an automated routine that might help volcanic observatories such as INGV-OE to detect (1) the starting and ending time of an LF episode, (2) the LF heights, (3) the erupted volumes, and (4) the TADR, saving time and especially providing consistency and uniform data extraction from thermal monitoring videos. This would allow a prompt understanding of the state of the volcano, and of the magnitude and intensity of each explosive paroxysm as soon as it ended, allowing a timely volcanic hazard assessment. In addition, both our automated routines, based on the CPD and Gaussian interpolation, proved to be reliable in constraining the climax phase of the paroxysm leading to a sustained eruptive column, which is the phase posing the greater hazard for its impact on aviation and the population. Conversely, the manual analysis had clear difficulties in distinguishing the threshold between the intermediate phase and the LF sustained phase [3,8,30].

However, in order to routinely use the algorithms proposed here, it is necessary to overcome some limits that we described earlier and illustrated in Figure 6. The first shortcoming arose from the detection of unwanted objects falling in the field of view of the eruption, such as the Sun or the surfaces it irradiated. In distinguishing this anomalous pattern, the Gaussian interpolation might help, which would reveal and remove any deviation from the normal trend. A more common problem, and one that is hard to handle, is the cloud interference, with clouds resulting from water droplets, ash or gases filtering or obscuring the thermal images [36–40]. In the cases of clouds partially obscuring the field of view, it was still possible to interpolate the missing data, provided that they represent a small percentage of the total duration of the episode, which was the procedure also carried out with manual analysis. However, when the cloud cover was too continuous and extended in many directions, such as the episode of 27 May 2021 (Table 2), there was no way to retrieve any useful data, and an estimation of the erupted volume can only be performed by considering the timing obtained from the seismicity or from borehole strainmeters [17,43–45,47,48], and multiplying this for the average TADR estimated for each single episode occurring during the whole period lasting from 13 December 2020 to 21 February 2022. Thus, considering for the episode # 32 of 27 May 2021, the duration of 60 min (=3600 s) estimated from the seismicity, and multiplying this time for the TADR averaged over the 65 episodes (Table 2; ~146 m$^3$ s$^{-1}$), we obtained an estimated volume of ~$0.53 \times 10^6$ m$^3$, which was in line with, and slightly below, the average of the other LF events here considered (Table 2). This brought the total erupted volume of pyroclastics or tephra, erupted between 13 December 2020 and 21 February 2022, to ~$65.2 \times 10^6$ m$^3$.

Considering the manually obtained results, from a volcanological point of view, it is worth noting that, if we excluded the two outliers of 23 and 31 March 2021, which emitted more than $4 \times 10^6$ m$^3$ of tephra (Table 2), the volume of pyroclastics erupted during the LF episodes, which occurred between mid-December 2020 and February 2022, increased with time (Figure 4), and also, the time-averaged discharge rate (TADR) increased with time (Figure 5). Figures 4 and 5 display a wide variability of values, and although this variability might hide shorter eruptive cycles, it is however clear that the trend of TADR and erupted volume increased with time. These observations, i.e., of the increase of erupted volume of pyroclastics and rate of eruption with time, would suggest that the sequence of paroxysmal

events was not yet over [31], and urges reliable and automated routines to be promptly developed, tested, and applied to the analysis of the LF episodes.

## 5. Conclusions

In this paper, we have presented the timing of start and end for 65 of the 66 LF episodes which took place at Etna volcano between 13 December 2020 and 21 February 2022, together with their duration, maximum and average LF heights, erupted volume of pyroclastics, and TADR (Table 2), obtained by manual analysis of the monitoring thermal images recorded by the INGV-OE network. We have then presented two automated routines, based on the CPD and Gaussian interpolation, that analyzed the thermal images and provided a fast and reliable way to obtain the same parameters acquired manually, in a timely way. The results obtained with the automated and manual routines are comparable (Figure 15), thus suggesting that a complete automation of the process is feasible. However, our analysis also highlighted important shortcomings arising from the presence of unwanted hot objects comprised in the field of view of the explosive episode that may lead to false results. Moreover, the presence of ash, weather, and gas clouds caused important interference with the data analysis, and might have reached the point of a complete obscuration of the field of view, as in the case of the episode #32 of 27 May 2021 (Table 2). We have shown that the Gaussian interpolation may limit the errors caused by a partial view, but more studies are necessary before this analysis can be routinely used for monitoring purposes. The results of our study showed an increasing magnitude (erupted volume) and intensity (TADR) of the explosive events in the period here considered (see Figures 4 and 5), and this issue would urge a faster and reliable analysis to be obtained as soon as possible, thus motivating the work presented here.

**Author Contributions:** Conceptualization, S.C. and G.N.; methodology, S.C. and G.N.; software, G.N.; validation, S.C. and G.N.; formal analysis, S.C. and G.N.; investigation, S.C. and G.N.; resources, S.C. and G.N.; data curation, S.C. and G.N.; writing—original draft preparation, S.C. and G.N.; writing—review and editing, S.C. and G.N.; visualization, S.C. and G.N.; supervision, S.C. and G.N.; project administration, S.C.; funding acquisition, S.C. All authors have read and agreed to the published version of the manuscript.

**Funding:** This research was funded by the Project FIRST—ForecastIng eRuptive activity at Stromboli volcano: timing, eruptive style, size, intensity, and duration, INGV-Progetto Strategico Dipartimento Vulcani 2019 (Delibera n. 144/2020; Scientific Responsibility: S.C.).

**Data Availability Statement:** The videos of eruptive activity used in this paper belong to the Istituto Nazionale di Geofisica e Vulcanologia, Osservatorio Etneo – Sezione di Catania and are used for monitoring purposes. Selected videos can be made available upon request to the first author of this paper.

**Acknowledgments:** We would like to thank the INGV-OE scientists and technicians for the monitoring network maintenance, and especially Michele Prestifilippo for providing information essential for this work. We thank Stephan Conway for revising the English style.

**Conflicts of Interest:** The authors declare no conflict of interest.

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
