# Peer review of "Comparison between Automated and Manual Detection of Lava Fountains from Fixed Monitoring Thermal Cameras at Etna Volcano, Italy"

_remotesensing, doi:10.3390/rs14102392_

Round 1

Reviewer 1 Report

Dear Authors

The overall quality of the manuscript is a good one, but I have a question:

In section 2 you wrote: “When visibility is limited to a few frames or interfering clouds are low and only partially obscure the lava fountain (LF), a linear interpolation is carried out on the data [3].”.

It wasn't clear to me how to determine if visibility was limited? Is it manual procedure?

Author Response

The overall quality of the manuscript is a good one, but I have a question:

In section 2 you wrote: “When visibility is limited to a few frames or interfering clouds are low and only partially obscure the lava fountain (LF), a linear interpolation is carried out on the data [3].”.

It wasn't clear to me how to determine if visibility was limited? Is it manual procedure?

We would like to thank the reviewer for his/her comments on our manuscript. We have clarified the above issue in the revised text explaining that visibility is determined by a manual procedure.

Reviewer 2 Report

The article is written quite well and I appreciated the fact that the Authors discussed in details all the limitations of their approach. The automatic routine they proposed can have important applications during a volcanic crisis. 

Comments:

The Authors should briefly discuss in the introduction the difference between the volume estimated from thermal images and the volume estimated from other data (e.g. field data, InSAR derived DEM etc.). This is important to have an idea of the error associated to the volumes reported in Table 2. I think that this information is available for the previous eruptions at Etna volcano.

In addition, in the section 3.4 (and maybe in the discussion section) the Authors should better quantify the differences of the three methods they used (Manual, CPD, Gauss). The visual comparison reported in figures 12-14 is important, but as for the main parameters (height, duration, volume and maybe TADR) would be equally important to quantify the difference (as a %). They can calculate the difference (as a %) for each episode and then they can write something like: “Volumes obtained with the manual method are from XX% to YY% higher than that obtained from the CPD and from XY% to YX% higher than that obtained with the Gaussian approach. On the contrary, the maximum difference in terms of volumes estimated with the CPD and the Gaussian approach is of XX% (episode X), while the minimum difference is of Y (episode Y).” the same for the height and duration.

Minor comments:

Abstract:

In the abstract the sentence “In this paper, we present an automatic routine that, when applied to thermal images, and providing good weather conditions, allowed us to detect” is not clear. Do you mean that this technique works only in good weather conditions?

Introduction.

The sentence “In particular, between 2011 and 2015, Etna produced more than 50 such eruptions [3,10-12], releasing a cumulative erupted volume of a similar order to a major flank eruption [2]”, should be better explained. It is not clear the message of this phrase. Are there differences between the summit and flank eruptions in terms of erupted volume?

The sentence “The volume erupted during a lava fountain (LF) episode quantifies the magnitude of the event, whereas the eruption rate is a function of its intensity [23].” could be rewritten as: “The volume erupted during a lava fountain (LF) episode quantifies the magnitude of the event, whereas the eruption rate determines its intensity”. I think it would be better because it is the intensity that is calculated from the eruption rate and not vice versa.

Methods

When you show the position of the INGV thermal monitoring cameras (Figure 1) maybe would be a good idea to show also the “looking direction” (or write a sentence saying that all the cameras look at the summit crater of the Etna).

As for the sentence “the corresponding area ??,=?,?, i.e. the actual number of pixels in the i-th region, returned as a scalar” please provide more details. Is ?? the corresponding area of what (total area of the white pixels?)? What is N (number of white pixels?)?

In the sentence: “?? and (??) being the horizontal and vertical coordinates, respectively” you have to put the round brackets at both (??) and (??) or you need to remove the round brackets to (??).

In Figure 2a would be better to add the colorbar.

Equation 1: w is not defined in the text. Please add what w is

Results

I agree with the Authors that the volume of pyroclastics erupted during the LF episodes increased with time, as shown in Figure 4, even though maybe there is also a kind of cyclicity, with periods in which the LF volume increases over the time followed by periods in which it decreases (that you might want to discuss). However, I think that this increase with time should be better shown on Figure 4 by adding a linear fitting or by trying working with the axes (you could see if by using a logarithmic y-axis the increase of the volume with the time becomes more evident).

3.2. Change Point Detection

Equation 4: please add what β is.

The sentence “the former is more suitable when due to external noise sources, e.g. poor visibility or interference by other kinds of hot objects, the automatic detection of the transition from Strombolian to paroxysmal activity and vice versa can be more problematic” does not work (at least one comma is missing after the word when). Maybe you can rewrite: “the former is more suitable when the automatic detection of the transition from Strombolian to paroxysmal activity and vice versa can be more problematic due to external noise sources (e.g. poor visibility or interference by other kinds of hot objects). 

Equation 2-4. Please quote from where you took them. 

3.3 Timing the lava fountains occurring at Etna during 2020-2022

What do you mean with the sentence: “the user can estimate quantities (e.g. the height and duration of the LF which are necessary for the calculation of the volumes of erupted material), also in this case with a considerable time saving”. Do you mean that the quantities are automatically estimated? Please better specify the meaning of this phrase.

Figure 13. What is represented on y-axis? Please make Figure 13 consistent with Figure 12 and 14 (in the legend man should be replaced with manual and the black line should be orange, right?).

Author Response

The article is written quite well and I appreciated the fact that the Authors discussed in details all the limitations of their approach. The automatic routine they proposed can have important applications during a volcanic crisis. 

We would like to thank the reviewer for his/her comments on our manuscript.

Comments:

The Authors should briefly discuss in the introduction the difference between the volume estimated from thermal images and the volume estimated from other data (e.g. field data, InSAR derived DEM etc.). This is important to have an idea of the error associated to the volumes reported in Table 2. I think that this information is available for the previous eruptions at Etna volcano.

We have now inserted the comparison with other data (strain for the total erupted volume, comprising pyroclastics and lava flows, and satellite for the lava flow proportion) and the estimated error of ~20% on the calculated volume of fluids, as stated by Bonaccorso and Calvari 2017.

In addition, in the section 3.4 (and maybe in the discussion section) the Authors should better quantify the differences of the three methods they used (Manual, CPD, Gauss). The visual comparison reported in figures 12-14 is important, but as for the main parameters (height, duration, volume and maybe TADR) would be equally important to quantify the difference (as a %). They can calculate the difference (as a %) for each episode and then they can write something like: “Volumes obtained with the manual method are from XX% to YY% higher than that obtained from the CPD and from XY% to YX% higher than that obtained with the Gaussian approach. On the contrary, the maximum difference in terms of volumes estimated with the CPD and the Gaussian approach is of XX% (episode X), while the minimum difference is of Y (episode Y).” the same for the height and duration.

We have added in the text the average error in percent and the standard deviation for the key parameter evaluated by using the automated approaches with respect to the manual ones.

Minor comments:

Abstract:

In the abstract the sentence “In this paper, we present an automatic routine that, when applied to thermal images, and providing good weather conditions, allowed us to detect” is not clear. Do you mean that this technique works only in good weather conditions?

Yes, because clouds may obscure completely visibility of the summit to the thermal images. We have now clarified this point in the abstract.

Introduction.

The sentence “In particular, between 2011 and 2015, Etna produced more than 50 such eruptions [3,10-12], releasing a cumulative erupted volume of a similar order to a major flank eruption [2]”, should be better explained. It is not clear the message of this phrase. Are there differences between the summit and flank eruptions in terms of erupted volume?

Yes, flank eruptions erupt an estimated volume ten times greater than summit eruptions, We have clarified this point and added a reference to Neri et al 2009.

The sentence “The volume erupted during a lava fountain (LF) episode quantifies the magnitude of the event, whereas the eruption rate is a function of its intensity [23].” could be rewritten as: “The volume erupted during a lava fountain (LF) episode quantifies the magnitude of the event, whereas the eruption rate determines its intensity”. I think it would be better because it is the intensity that is calculated from the eruption rate and not vice versa.

Thanks a lot, done

Methods

When you show the position of the INGV thermal monitoring cameras (Figure 1) maybe would be a good idea to show also the “looking direction” (or write a sentence saying that all the cameras look at the summit crater of the Etna).

Thanks a lot for this suggestion. We have added this sentence at the start of the methods.

As for the sentence “the corresponding area ??,=?,?, i.e. the actual number of pixels in the i-th region, returned as a scalar” please provide more details. Is ?? the corresponding area of what (total area of the white pixels?)? What is N (number of white pixels?)?

We have clarified that represents the area in pixel of each recognized object, represented in the binarized image in white color and that N is the number of recognized objects.

In the sentence: “?? and (??) being the horizontal and vertical coordinates, respectively” you have to put the round brackets at both (??) and (??) or you need to remove the round brackets to (??).

We have removed the brackets

In Figure 2a would be better to add the colorbar.

The color bar was not included in the original figure, but we added in the figure caption the values of temperature displayed by the colours.

Equation 1: w is not defined in the text. Please add what w is

We have added that is the normalized area of the i-th object.

Results

I agree with the Authors that the volume of pyroclastics erupted during the LF episodes increased with time, as shown in Figure 4, even though maybe there is also a kind of cyclicity, with periods in which the LF volume increases over the time followed by periods in which it decreases (that you might want to discuss).

In order to consider cyclicity it would be necessary to consider time spans much longer than those that we took into account here. However, we have modified the text considering also this possibility

However, I think that this increase with time should be better shown on Figure 4 by adding a linear fitting or by trying working with the axes (you could see if by using a logarithmic y-axis the increase of the volume with the time becomes more evident).

We have modified the figures 4 and 5 adding the linear fitting and its equations

3.2. Change Point Detection

Equation 4: please add what β is.

We have added that β represents a positive coefficient which weights the number of searched change points.

The sentence “the former is more suitable when due to external noise sources, e.g. poor visibility or interference by other kinds of hot objects, the automatic detection of the transition from Strombolian to paroxysmal activity and vice versa can be more problematic” does not work (at least one comma is missing after the word when). Maybe you can rewrite: “the former is more suitable when the automatic detection of the transition from Strombolian to paroxysmal activity and vice versa can be more problematic due to external noise sources (e.g. poor visibility or interference by other kinds of hot objects). 

We changed the sentence as suggested by the reviewer, whom we thank

Equation 2-4. Please quote from where you took them. 

We quoted from where we learn this equation.

3.3 Timing the lava fountains occurring at Etna during 2020-2022

What do you mean with the sentence: “the user can estimate quantities (e.g. the height and duration of the LF which are necessary for the calculation of the volumes of erupted material), also in this case with a considerable time saving”. Do you mean that the quantities are automatically estimated? Please better specify the meaning of this phrase.

We have clarified that the meaning of the sentence is that the user can speed up the computation of key quantities.

Figure 13. What is represented on y-axis? Please make Figure 13 consistent with Figure 12 and 14 (in the legend man should be replaced with manual and the black line should be orange, right?).

The label in the y-axis expresses the duration in minutes. We have updated the Figure 13 and now is consistent with Fig 12 and 14. The y-axis was also clarified represents the duration in minutes.

Reviewer 3 Report

This is a good manuscript that deserves to be published.

  • I suggest some modifications on Introduction topic. There several discussions that should be presented in the appropriate topic.
  • Send the paper to a native English-speaker for final revision. 

See attached PDF file.

Author Response

This is a good manuscript that deserves to be published.

We would like to thank the reviewer for his/her comments on our manuscript.

I suggest some modifications on Introduction topic. There several discussions that should be presented in the appropriate topic.

Send the paper to a native English-speaker for final revision.

The paper was reviewed by a native English speaker.

See attached PDF file.

We changed automatic into automated both in the title and everywhere in the manuscript, and changed the order of keywords following the sequence of the title, and thank Rev. 2 for this correction.

At page 2 the reviewer suggested to use volcaniclastic instead than pyroclastics, but we did not change this word because volcaniclastic comprises also reworked products, whereas pyroclastics refers only to primary eruptive products fragmented upon eruption, which was the material erupted during the lava fountains.